# The Roles of Carbohydrate Response Element Binding Protein in the Relationship between Carbohydrate Intake and Diseases

**DOI:** 10.3390/ijms222112058

**Published:** 2021-11-08

**Authors:** Katsumi Iizuka

**Affiliations:** 1Department of Clinical Nutrition, Fujita Health University, Toyoake 470-1192, Japan; katsumi.iizuka@fujita-hu.ac.jp; 2Yutaka Seino Distinguished Center for Diabetes Research, Kansai Electric Power Medical Research Institution, Kobe 650-0047, Japan

**Keywords:** glucose, carbohydrate response element binding protein, ChREBP, fructose, type 2 diabetes mellitus, T2DM, high-fructose corn syrup, HFCS, sugar-sweetened beverages, SSBs

## Abstract

Carbohydrates are macronutrients that serve as energy sources. Many studies have shown that carbohydrate intake is nonlinearly associated with mortality. Moreover, high-fructose corn syrup (HFCS) consumption is positively associated with obesity, cardiovascular disease, and type 2 diabetes mellitus (T2DM). Accordingly, products with equal amounts of glucose and fructose have the worst effects on caloric intake, body weight gain, and glucose intolerance, suggesting that carbohydrate amount, kind, and form determine mortality. Understanding the role of carbohydrate response element binding protein (ChREBP) in glucose and lipid metabolism will be beneficial for elucidating the harmful effects of high-fructose corn syrup (HFCS), as this glucose-activated transcription factor regulates glycolytic and lipogenic gene expression. Glucose and fructose coordinately supply the metabolites necessary for ChREBP activation and de novo lipogenesis. *Chrebp* overexpression causes fatty liver and lower plasma glucose levels, and ChREBP deletion prevents obesity and fatty liver. Intestinal ChREBP regulates fructose absorption and catabolism, and adipose-specific *Chrebp*-knockout mice show insulin resistance. ChREBP also regulates the appetite for sweets by controlling fibroblast growth factor 21, which promotes energy expenditure. Thus, ChREBP partly mimics the effects of carbohydrate, especially HFCS. The relationship between carbohydrate intake and diseases partly resembles those between ChREBP activity and diseases.

## 1. Introduction

Most foods (e.g., bread, beans, rice, and milk) contain carbohydrates, which are macronutrients in the three groups monosaccharides, disaccharides, and polysaccharides [1]. In addition to the amount of carbohydrates, the type of carbohydrate consumed is essential for the prevention of several diseases, such as cardiovascular diseases and type 2 diabetes mellitus (T2DM). In particular, overeating foods rich in sucrose and fructose have been considered to induce obesity and T2DM [2]. Compared with glucose, fructose is more rapidly converted into triglycerides and causes fatty liver and body weight gain [2]. The capacity of fructose to produce advanced glycation end products, which are generated through nonenzymatic glycation in tissues and thereby alters tissue functions, is 10 times greater than that of glucose [3,4,5]. Thus, fructose is considered to cause diabetic complications much more easily than glucose. Moreover, many studies have shown that the intake of fructose-containing sugar-sweetened beverages (SSBs) is positively associated with weight gain and the risk of T2DM [6,7].

Research on the mechanism by which fructose and sucrose induce fatty liver and body weight gain has focused on the role of carbohydrate response element binding protein (ChREBP) [5,8,9]. ChREBP is a transcription factor that regulates de novo lipogenesis through lipogenic gene expression [10]. Glucose- and fructose-derived metabolites activate ChREBP [11,12,13,14], which regulates glucose and fructose metabolism through gene expression in the intestine and liver [10,15,16,17]. ChREBP also regulates fibroblast growth factor 21 (FGF21), a hepatokine that suppresses sugar consumption and promotes energy expenditure [18,19,20]. Therefore, ChREBP underlies the effects of carbohydrates on glucose and lipid metabolism.

In this review, I try to clarify the role of ChREBP in the relationship between carbohydrate intake and diseases. I first describe the relationship between carbohydrate intake and mortality, then the relationship between ChREBP and nutrition-related diseases, and finally unsolved problems related to ChREBP and nutrition.

## 2. Carbohydrate and Diseases

### 2.1. Carbohydrate Intake and Mortality

A recent study showed a nonlinear relationship between carbohydrate intake and mortality [21,22,23]. High- and low-carbohydrate diets were associated with reversal when the energy from carbohydrates was replaced with plant-derived protein or fat [21]. Similarly, some reported that higher carbohydrate intake was associated with a higher risk of mortality [22]. These results suggested that replacement of carbohydrates (e.g., rice and bread) with plant-based protein or fat (e.g., tofu) may be beneficial for lowering mortality. Some reported that carbohydrate intake showed a nonlinear association with mortality; there was no association at 20–50% of total energy intake but a positive association at 50–70% of energy intake [23]. Notably, only sugar intake resembled a J-shaped pattern [23]. Thus, overeating carbohydrates is associated with a higher risk of mortality, and this relationship is probably influenced by added sugars (Figure 1).

As the amount of mono- and disaccharides increases, mortality also increases. As sugar content increases in foods, the risk of obesity, T2DM, NAFLD, and mortality worsen in humans. The Y axis indicates the percentages of starch, glucose, fructose, sucrose, and dietary fiber to total carbohydrate contents in foods (rice, bread, corn, orange, orange juice, soda, and high-fructose corn syrup). Abbreviation: high-fructose corn syrup, HFCS.

### 2.2. Sucrose, Fructose, and Dietary Fibers

#### 2.2.1. Sucrose and Fructose

Among the several types of carbohydrates, sucrose and fructose are considered candidates that increase cardiometabolic risk [24,25,26]. Sucrose intake quickly induces de novo lipogenesis [27,28]. Glucokinase, phosphofructokinase, and liver-type pyruvate kinase are rate-limiting enzymes in glycolysis that slow glucose flux [29]. In contrast, there are no rate-limiting enzymes in fructolysis, and acute fructose infusion sometimes causes lactic acidosis [30]. Therefore, fructose is easier to metabolize in the fructolytic and glycolytic pathways (Figure 2). Glyceraldehyde is converted into glycerol and used for triglyceride synthesis. Acetyl-CoA can be used as the substrate for de novo lipogenesis. Moreover, fructose produces 10-fold more glycation end products than glucose [3]. Therefore, fructose has been considered harmful regarding the development of insulin resistance and fatty liver.

However, intestinal fructose absorption is slower than glucose absorption [31]. Consistently, oral fructose injection did not cause postprandial hyperglycemia [15]. The intestinal absorption of fructose is much slower and more complex than that of glucose. Interestingly, recent studies showed that smaller doses of fructose were metabolized into glucose and its metabolites in the intestine [32], whereas when administered at higher doses (1 g/kg body weight), fructose was delivered to the liver and metabolized into fatty acids and triglycerides [32]. These results suggested that the intestine is a barrier for fructose flux into the liver to prevent lactic acidosis and fatty liver. Regarding unabsorbed fructose, gut microbiota ferment fructose into short-chain fatty acids. In patients with fructose malabsorption, fructose intake leads to osmotic diarrhea as well as gas and bloating due to fermentation in the colon. These findings are consistent with the result that fructose consumption was associated with irritable bowel syndrome in some patients with fructose malabsorption [33].

Sucrose and glucose/fructose feeding elicited insulin resistance and fatty liver in animal models [34,35]. In contrast, some studies have reported that only fructose feeding failed to induce excess weight gain in mice [36]. Moreover, some meta-analyses of controlled human intervention studies have failed to demonstrate adverse glycemic effects unique to fructose in the diet in people with diabetes [37]. A critical study on the relationship between metabolic phenotypes and the glucose-to-fructose ratio in food was recently reported [38]. Energy intake and adiposity were highest after consuming a diet containing a 50:50 ratio of glucose to fructose (similar to HCSF) and a low protein content (LP and HFCS diet) [38]. Although energy expenditure by FGF21 was highest on the LP and HFCS diets, this diet evoked worse insulin resistance, glucose tolerance, and liver fat accumulation [38]. These changes observed with the LP and HFCS diets were more remarkable than those with the LP and sucrose diets [38]. These results suggested that high-fructose corn syrup is more harmful than sucrose and that glucose and fructose cooperatively promote the development of fatty liver.

High-fructose corn syrup (HFCS) is a sweetener made from corn starch [39] that is further processed by D-xylose isomerase, which converts some glucose to fructose. HFCS is used in processed foods, breakfast cereals, soft drinks, and filling jellies [39]. Some meta-analyses of prospective cohort studies and RCTs have shown that SSB consumption promotes weight gain and obesity in children and adults [6,7,40]. Moreover, SSB consumption is associated with a significantly elevated risk of T2DM, whereas the association between artificially sweetened beverages and T2DM was primarily explained by health status, pre-enrollment weight change, dieting, and body mass index [41]. A meta-analysis of four observational studies found a significant positive association between higher SSB consumption and nonalcoholic fatty liver disease (NAFLD) in both men and women [42]. Moreover, all-cause mortality was higher among participants who consumed two or more glasses per day of all soft drinks, sugar-sweetened soft drinks, or artificially sweetened soft drinks than among those who consumed <1 glass per month [43]. Positive associations were observed between artificially sweetened soft drinks and death from cardiovascular disease and between sugar-sweetened soft drinks and death from digestive diseases [43]. Collin et al. evaluated the associations of SSBs and 100% fruit juices with coronary heart disease (CHD) mortality and all-cause mortality [44,45]. With each additional 12 oz of SSBs or fruit juice alone, the risk-adjusted all-cause mortality hazard ratios (HRs) were 1.11 (95% CI, 1.03–1.19) and 1.24 (95% CI, 1.09–1.42), respectively [44,45]. In contrast, participants who consumed 10% or more of their daily calories as SSBs tended to have a nonsignificant increase in CHD mortality [44,45]. Therefore, in discussing the association between carbohydrates and mortality, we should not overeat carbohydrates, especially sugars.

#### 2.2.2. Dietary Fiber

Dietary fiber is composed of a spectrum of nondigestible food ingredients, including nonstarch polysaccharides, oligosaccharides, lignin, and analogous polysaccharides [46]. Dietary fiber promotes a reduction in the contact time of carcinogens and promotes a “healthy” gut microbiota because certain dietary fibers are fermentable [46]. Many of the health benefits can be attributed to the fermentation of dietary fiber into short-chain fatty acids (SCFAs) (acetate, propionate, and butyrate) in the colon. The SCFAs released in the intestinal lumen are readily absorbed and used as energy sources by colonocytes, the liver, and muscle. Moreover, SCFAs play an important role as modulators of immunological substances and thereby suppress chronic inflammation [46]. Thus, dietary fiber decreases intestinal cholesterol uptake, lowers blood pressure, and improves insulin resistance and anti-inflammatory effects. In addition to the anti-inflammatory and metabolic effects of dietary fiber, dietary fiber is associated with lower mortality. In a human study, dietary fiber was inversely associated with all-cause mortality [23,47]. Total, soluble, and insoluble fiber intake were inversely associated with all-cause mortality [47,48].

Increased quintiles of dietary fiber intake were significantly associated with decreased mortality due to total cardiovascular disease, respiratory disease, and injury [48]. Total fiber intake was significantly inversely associated with cancer mortality in Europe [49]. Interestingly, fiber from fruits, beans, and vegetables, but not that from cereals, was inversely associated with total mortality [49]. Similarly, whole grains rich in dietary fibers also have different effects than refined grain with regard to mortality. Whole grain intake is inversely associated with mortality. In contrast, refined grains lack one or more of the three key parts of grain (bran, germ, or endosperm) and contain lower amounts of dietary fiber. Consistently, refined grain intake was not associated with mortality [50]. These results also suggest that food form may affect mortality. The differences in soluble and insoluble fiber composition between foods may affect mortality.

### 2.3. ChREBP and Diseases

#### 2.3.1. What Is ChREBP?

As described in Section 2.2, excess intake of fructose and sucrose can worsen body weight and glycemic control. To understand the harmful effects of HFCS, it is essential to appreciate the role of ChREBP in de novo lipogenesis from carbohydrates. During feeding, plasma glucose and insulin levels are increased, and both glucose and insulin promote the conversion of excess carbohydrates into triglycerides through effects on lipogenic gene expression [51,52]. Therefore, chronic overeating of carbohydrates causes liver and adipose triglyceride accumulation and insulin resistance. Glucose and insulin activate the transcription factors ChREBP and SREBP1c, respectively [51].

Formerly, carbohydrate response elements in the promoters of glucose response genes, such as liver-type pyruvate kinase, fatty acid synthase, and acetyl-CoA carboxylase, were identified as two E boxes separated by five nucleotide spaces (CAYGYGnnnnnCRCRTG); however, transcription factors that bind carbohydrate response elements (ChoREs) in lipogenic genes remain unidentified [8,53,54]. Yamashita and Uyeda et al. identified ChREBP as a transcription factor that binds to ChoRE. In the livers of sucrose-fed rats, the DNA binding of ChREBP to ChoRE in LPK increased, and cotransfection assays using a reporter vector with the LPK promoter and ChREBP expression vector revealed that ChREBP induces glucose response gene transcription. By sequence analysis, ChREBP was identified as one of 17 genes deleted in Williams–Beuren syndrome, which is a neurodevelopmental disorder characterized by heart and vascular disease, dysmorphic facial features, and intellectual disability [8,54]. ChREBP and Mlx form a heterodimer and bind to the carbohydrate response element [55,56,57]. Therefore, ChREBP is also called WBSCR17 or mlx interacting protein like (Mlxipl).

#### 2.3.2. The Regulation of ChREBP Activity

ChREBP is a large protein (Mr = 94,600 Da) composed of 864 amino acids and belongs to the MONDO family of basic helix-loop-helix- and leucine zipper-type transcription factors [6]. ChREBP regulates lipogenic, glycolytic, fructolytic, gluconeogenic, and hormone/receptor pathways [8,9,10,52,57,58] and is expressed in the liver, intestine, muscle, white and brown adipose tissue, pancreatic islets, and adrenal glands [10,54,59]. ChREBP has two isoforms, ChREBP alpha and ChREBP beta [60]. ChREBP alpha has a nuclear localization signal (NLS), nuclear export signal (NES), and glucose-sensing module composed of a low glucose inhibitory domain (LID) and conserved glucose-response activation element [61,62,63]. The transcriptional activity of ChREBP alpha is regulated by glucose levels. In contrast, ChREBP beta lacks NLS, NES, and LID; is localized in the nucleus; and is constitutively active even at low glucose levels [60]. As ChREBP beta is transcribed from an alternative first exon 1b promoter containing ChoRE, ChREBP alpha induced ChREBP beta through DNA binding of ChREBP to ChoRE. In contrast, ChREBP beta suppresses ChREBP alpha expression. Thus, ChREBP alpha and ChREBP beta constitute feed-forward and feedback loops [64,65]. Regarding the regulation of ChREBP transcription activity, phosphorylation/dephosphorylation, GlcNacylation, acetylation, and the allosteric effects of metabolites are involved [11,12,13,14]. Moreover, other transcription factors, such as HNF4a and FXR, can modify ChREBP transcriptional activity through interactions [66,67], and LXR can induce *Chrebp* mRNA in mice.

Regarding the regulation of ChREBP, glucose-derived metabolites are important. Sucrose and fructose are more potent inducers of lipogenesis than starch and glucose. It is easier to metabolize fructose than glucose into acetyl-CoA, a substrate for lipogenesis [30,68]. A high-sucrose diet increased both xylulose-5-phosphate (Xu-5-P) and fructose-2,6-bisphosphate (F-2,6-P), which are activators of ChREBP [69]. F-2,6-P also activates PFK2 and thereby PFK, a rate-limiting glycolytic enzyme [70]. Glucose-derived metabolites (G6P, Xu-5-P, and F-2,6-P_2_) activate ChREBP and thereby induce gene expression in the fructolytic pathway [11,12,13,14]. These substrates are considered to contribute to ChREBP activation via ChREBP dephosphorylation (F-2,6-P and Xu-5-P) and likely via conformational changes (G6P) [11,12,13,14]. Moreover, ChREBP O-GlcNAcylation also contributes to ChREBP activity [71,72,73]. UDP-GlcNAc is a glucose-derived metabolite in the hexosamine biosynthesis pathway and is used as a substrate for ChREBP O-GlcNAcylation [71,72,73]. Unlike glucose metabolism, fructose is much more rapidly converted into acetyl-CoA due to a lack of rate-limiting enzymes. Therefore, the potency of fructose alone on ChREBP may be much weaker than that of glucose, but under real-world conditions, both fructose and glucose are ingested at the same time in many cases. In the presence of glucose, intestinal fructose absorption is increased compared with that of fructose alone [31]. Therefore, glucose-derived metabolites mainly activate ChREBP, which then induces fructolysis and de novo lipogenesis through gene expression (Figure 2). Thus, sucrose and HFCS more potently induce lipogenic gene expression than fructose or glucose alone, and ChREBP potently regulates lipogenesis upon modulation by both glucose and fructose (Figure 3).

HFCS contains equal amounts of glucose and fructose and promotes intestinal fructose absorption. Much of the ingested fructose is converted into glucose metabolites in the intestine and delivered to the liver. HFCS also induces lipogenic genes and VLDL secretion. Hepatic ChREBP induces FGF21, a suppressor of sweet consumption, and adipose ChREBP regulates glucose uptake, lipogenesis, and branched chain fatty acid synthesis.

The mechanism of ChREBP activity suppression has also been reported [54,74,75,76,77]. Glucagon is a hormone known to increase hepatic glucose output and thereby increase plasma glucose levels. Glucagon increases cAMP levels and thereby suppresses ChREBP transcription activity through promotion of ChREBP phosphorylation [54,74]. Moreover, AMP is also an important modulator that suppresses ChREBP activity [75,76,77]. AMP is a metabolite that is increased during starvation, and AMP-activated protein kinase (AMPK) phosphorylates ChREBP and then inactivates ChREBP transcription activity [75]. Moreover, AMP also suppresses ChREBP transcription activity as an allosteric inhibitor [76,77]. Thus, under starvation conditions, ChREBP activity is shut off by phosphorylation and allosteric inhibition.

#### 2.3.3. The Role of ChREBP in Several Tissues

Regarding its physiological roles, ChREBP regulates glucose and lipid metabolism in various tissues. In the liver, hepatic *Chrebp* overexpression evoked fatty liver [78,79], and the suppression of ChREBP in obese mice prevented body weight gain and fatty liver [80,81]. Interestingly, ChREBP overexpression and suppression improved glucose tolerance and lowered plasma triglyceride levels in lean and obese mice, respectively [78,79,80,81]. Regarding ChREBP overexpression, increased FGF21 levels improve glucose tolerance and lower plasma triglyceride levels in mice [78,79]. Furthermore, by ChREBP suppression, the decreased appetite and body weight loss partly improved glucose tolerance, and decreased TAG synthesis lowered plasma triglyceride levels [80,81]. Thus, ChREBP activity is positively associated with liver lipid content, but ChREBP exerts a U-shaped effect on glucose tolerance and plasma lipid levels in different manners.

In the intestines, ChREBP regulates both hepatic and fructose metabolism [15,16,17]. Interestingly, individuals deficient in ChREBP cannot eat a high-sucrose and -fructose diet, and ChREBP deficiency leads to death [10]. When a high-fructose diet was consumed, intestinal ChREBP inhibition caused irritable bowel syndrome [15]. Moreover, a sucrose isomaltase inhibitor and moderate amounts of sucrose feeding also caused irritable bowel syndrome due to increased growth of gut microbiota [15]. These results suggested that unabsorbed fructose promotes osmotic diarrhea and increases gut microbiota fermentation. Low FODMAP (fermentable oligo-, di-, monosaccharides, and polyols) diets are recommended for patients with irritable bowel syndrome [82]. In patients with fructose malabsorption, lowering fructose intake is sometimes effective for the treatment of irritable bowel syndrome. These results are consistent with the findings that fructose and sucrose consumption cause irritable bowel syndrome in ChREBP^-/-^ mice. Whether dietary fiber promotes irritable bowel syndrome in ChREBP^-/-^ mice remains unclear. As dietary fiber also promotes gut microbiota fermentation, dietary fiber intake may also cause irritable bowel syndrome in these mice. I will further investigate this phenomenon.

ChREBP is abundantly expressed in white adipose tissue [10,54]. ChREBP is not expressed in preadipocytes, but it is induced during adipogenesis [83]. ChREBP consistently regulated lipogenic gene and Glut4 expression in adipose tissues [84]. Importantly, Barbara Kahn’s group reported that the absence of *Chrebp* caused systemic insulin resistance by reducing the levels of palmitic acid esters of hydroxy stearic acids (PAHSAs) [84]. Moreover, the administration of PAHSAs improved insulin sensitivity and glucose intolerance through muscle glucose uptake and hepatic glucose output in high-fat-fed mice [85]. Whether PAHSA modulates ChREBP transcription activity remains unclear, but this is a very important finding indicating that PAHSAs may be important molecules in regulating insulin sensitivity. PAHSA will be both an important therapeutic target for diabetes mellitus and a biological marker for insulin sensitivity.

ChREBP is also highly expressed in brown adipose tissue [65,86,87,88]. ChREBP is induced by lipogenic genes during adipogenesis, and Chrebp deletion causes decreased lipogenic gene expression [86]. Recently, ChREBP activation was shown to suppress thermogenesis by increasing lipid droplets and decreasing mitochondrial content in brown adipose tissue [87]. The increased conversion of acetyl-CoA into malonyl CoA likely suppressed lipolysis. UCP-1 and Dio2 levels were consistently decreased in mice with brown adipose tissue-specific ChREBP overexpression [87]. Mitochondrial cristae were decreased in brown adipose tissues from ChREBP knockout mice [88]. Inconsistently, some studies have reported that T3 and glucose coordinately stimulate ChREBP-mediated Ucp1 expression in brown adipocytes from male mice [89]. Thus, ChREBP overexpression acts as a negative regulator of thermogenesis in brown adipocytes, but *Chrebp* suppression may also disrupt thermogenesis in a different manner. ChREBP may also exhibit a U-shaped regulatory effect on the function of brown adipose tissues.

ChREBP is moderately expressed in the adrenal glands [59]. Plasma cholesterol levels are also decreased in *Chrebp*^-/-^ mice. Moreover, cholesterol is converted into cortisone in adrenal glands. *Chrebp*^-/-^ mice showed normal adrenal cholesterol and cortisol content but decreased triglyceride content [59]. Therefore, ChREBP may play a role in adrenal lipogenesis but not steroidogenesis [59].

ChREBP is highly expressed in pancreatic β cells [90,91,92,93]. Moreover, ChREBP expression is increased in obese mice. ChREBP overexpression consistently caused glucose-induced islet proliferation [90]. These results suggest that ChREBP plays a role in glucose-induced β cell proliferation [91,92,93]. Whether ChREBP deletion suppresses β cell proliferation remains unclear. mTOR suppresses the formation of the ChREBP/Mlx heterodimer complex, and insulin activates both the mTOR pathway and glucose metabolites, which stimulate ChREBP activity [93]. mTOR suppresses ChREBP activity, while increased glucose metabolites increase ChREBP activity [93]. Whether the potency of mTOR is superior to that of glucose metabolites remains unclear. In the kidneys, too, ChREBP is expressed. Because ChREBP expression is observed mainly in the proximal renal tube [94], *Chrebp* deletion causes improvement in proteinuria in streptozotocin-induced diabetic mice [95,96]. As one possible mechanism, ChREBP deficiency may alleviate diabetes-associated renal lipid accumulation by inhibiting mTORC1 activity [96].

ChREBP potentially regulates the intake of foods, especially sweets. Hepatokine FGF21 is regulated by ChREBP at the transcriptional level [18,19,20]. In rodents, FGF21 reduced sweet consumption, whereas FGF21 suppression increased sugar consumption [97,98]. In humans, plasma FGF21 levels increased acutely after oral sucrose ingestion and were elevated in fasted sweet-disliking individuals. These results suggest that the glucose/fructose-ChREBP-FGF21 pathway forms a negative feedback loop regulating sugar-eating behavior [99].

Finally, the relationship between ChREBP and malignant tumors has been investigated [100,101,102]. In malignant tumors, such as colon cancer, hepatocellular carcinoma, and prostate cancer, ChREBP levels are positively correlated with cancer mortality [100,101,102]. This may occur because ChREBP plays an important role in regulating tumor cell proliferation [102]. Suppression of *Chrebp* in vitro and in vivo leads to a p53-dependent reduction in tumor growth [88], which is consistent with evidence that ATP citrate inhibitors suppress tumor cell proliferation [103].

Thus, these results indicate that ChREBP has both beneficial and harmful effects on the development of chronic diseases, such as metabolic syndrome and cancers (Figure 4). These findings are consistent with results found in humans in that carbohydrate intake is nonlinearly associated with mortality [21,22,23].

*Chrebp* suppression prevents obesity and fatty liver but promotes sucrose insufficiency and insulin resistance. *Chrebp* overexpression promotes lower plasma glucose but fatty liver.

## 3. Future Perspective

Studies in humans have shown that carbohydrate type is an essential determinant of food intake. Moreover, the different effects of whole fruits and sugar-sweetened soft drinks are interesting. In an animal study, the route of glucose/fructose administration was not considered. Therefore, to evaluate the effect of carbohydrates on metabolic conditions, we should consider the composition of carbohydrates (glucose, fructose, sucrose, starch, and dietary fiber) (Figure 5). Considering that the ratio of glucose to fructose determines energy intake, energy expenditure, and liver triglyceride accumulation, it is rational to predict that this ratio will also govern ChREBP activity; we will try to clarify this hypothesis in future studies. The association between dietary fiber and ChREBP remains unclear. The gut microbiota converts dietary fiber into short-chain fatty acids, which are delivered to the liver. Acetylated ChREBP readily binds to ChREBP-response elements and increases lipogenesis-related gene expression in the liver [104]. Acetate itself is converted into acetyl-CoA and used as a substrate for lipogenesis. Although gut microbiota-derived acetate contributes to lipogenesis, further investigation of whether dietary fiber intake might be associated with ChREBP activity is needed (Figure 5).

The order of eating (vegetable first and carbohydrate last) is also known as the method of lowering plasma glucose and body weight [105,106,107]. Whether the order of eating (vegetable first and carbohydrate last) can affect ChREBP activity remains unclear. Lowering postprandial glucose and insulin levels may suppress ChREBP activity at adequate levels. In addition to plasma glucose and insulin levels, the order of eating can affect GIP and GLP-1 secretion [107]. Whether ChREBP affects the secretion and effect of GIP and GLP-1 secretion is an interesting topic.

Currently, a low-carbohydrate diet is widely recommended to obese people for the purpose of lowering body weight. In elderly patients, protein intake is recommended to prevent sarcopenia [108]. However, excess protein intake decreases fat and carbohydrate intake and thereby total energy intake. Specifically, protein content is an essential determinant of food intake; this protein leverage hypothesis states that small and large amounts of protein intake cause overconsumption and underconsumption, respectively, of fats and carbohydrates [109,110]. Moreover, excess carbohydrate restriction promotes muscle degradation and increases the risk of sarcopenia. As carbohydrates are unquestionably more efficient at protein sparing than fats, additional carbohydrate restriction will be harmful [111,112]. In other words, as insulin promotes protein synthesis and glucose depletion promotes protein degradation, balancing adequate ratios of carbohydrates, fat, and protein is important to prevent sarcopenia.

The role of ChREBP in muscle function is also an interesting topic.

The relationship between ChREBP and muscle function, MondoA, a ChREBP paralog, has been studied. MondoA, a ChREBP paralog, is abundantly expressed in muscle [113,114]. MondoA is also activated by glucose and fructose [115,116]. MondoA suppresses insulin signaling and thereby reduces glucose uptake through TXNIP and ARRDC4 expression [113]. Mice with muscle-specific MondoA deficiency were partially protected from insulin resistance and muscle TAG accumulation in the context of diet-induced obesity [113]. Why MondoA suppresses glucose uptake even though MondoA is activated by glucose seems contradictory. In fact, some studies have reported that muscle-specific *MondoA**^-/-^* mice exhibit muscle fiber atrophy, a reduced proportion of type II fibers compared to type I fibers, and an increased muscle glycogen level [114]. These results suggested that both increasing and decreasing MondoA activity may promote muscle dysfunction in a different manner.

Regarding ChREBP, *Chrebp*^-/-^ mice show insulin resistance [10]. Unlike that in *MondoA^-/-^* mice, the muscle glycogen content in *Chrebp*^-/-^ mice is similar to that in wild-type mice [10]. However, as insulin resistance may accelerate muscle protein degradation [117], ChREBP suppression might promote protein degradation and sarcopenia. In *Mondoa*^-/-^ mice, ChREBP might compensate to regulate glucose metabolism and improve muscle glucose uptake; however, the MondoA binding site (CACGTG) was not always the same as the ChREBP binding site (CAYGYGnnnnnCRCRTG) [6,57,115,116]. To solve the problem of whether ChREBP or MondoA is a dominant factor that regulates glucose metabolism in muscle, the development of *Chrebp^-/-^* x. *Mondoa^-/-^* mice will be needed.

## 4. Conclusions

Recent human studies have shown that the appropriate amount of carbohydrate intake is essential for longevity. These findings are the same as the results of studies using mouse models modulating ChREBP activity. Excess carbohydrate restriction or carbohydrate (especially sugar) intake may be harmful through the overmodification of ChREBP activity. Clarifying the position of ChREBP in diseases and diet therapy will be important in understanding the relationship between carbohydrate intake and mortality.

## Figures and Tables

**Figure 1 ijms-22-12058-f001:**
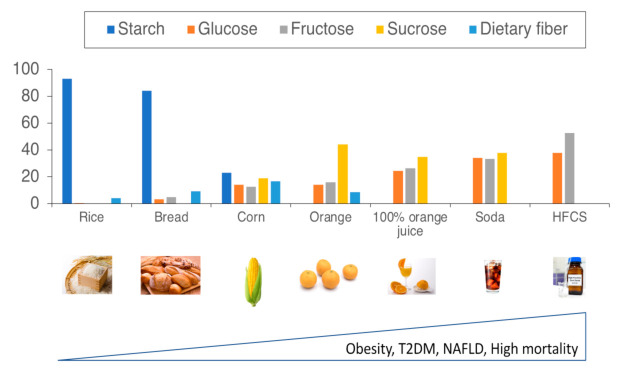
The association between carbohydrate type and mortality.

**Figure 2 ijms-22-12058-f002:**
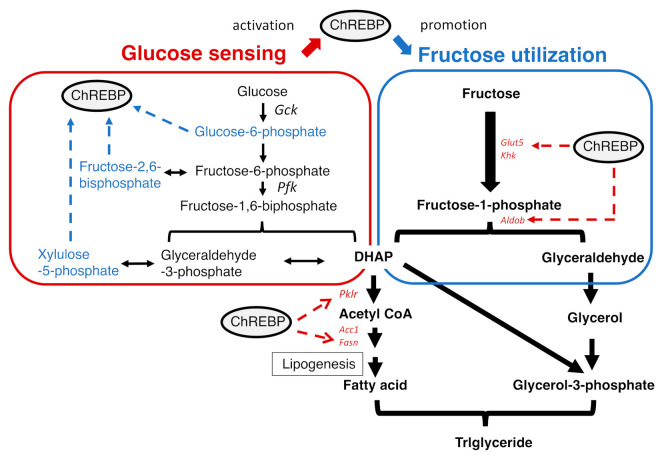
Glucose activates ChREBP, which then promotes fructolysis. Glucose increases the levels of metabolites that activate ChREBP. Upon activation, ChREBP induces fructolytic and lipogenic gene expression.

**Figure 3 ijms-22-12058-f003:**
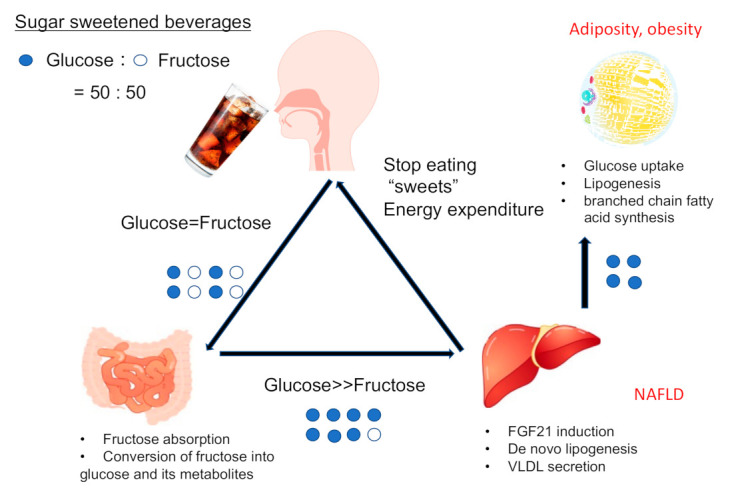
High-fructose corn syrup and ChREBP.

**Figure 4 ijms-22-12058-f004:**
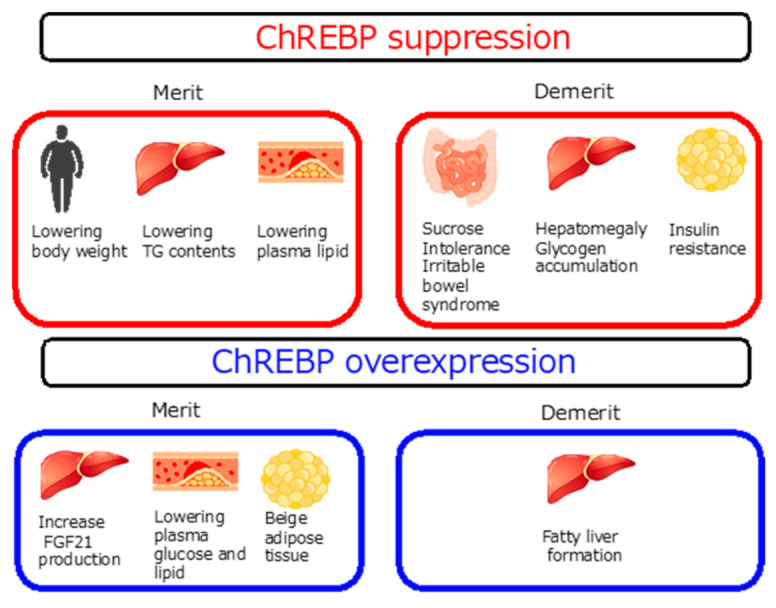
Merit and demerit of modulating ChREBP activity.45.

**Figure 5 ijms-22-12058-f005:**
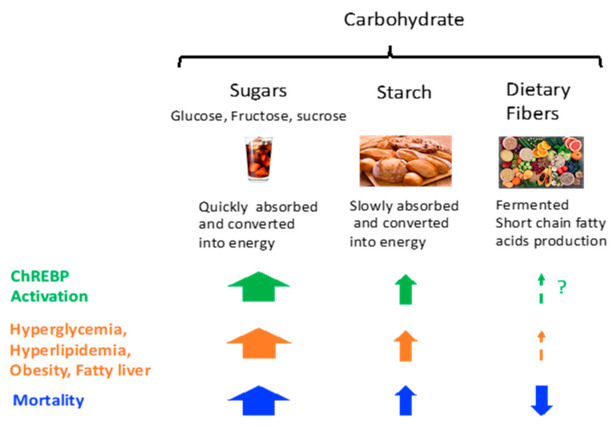
Carbohydrate intake and ChREBP activity, metabolic syndrome, and mortality. Sugars potently activate, starch modestly activates, and dietary fiber probably weakly activates ChREBP. The relationship between carbohydrate intake and ChREBP activity is similar to those of metabolic syndrome (hyperglycemia, hyperlipidemia, obesity, and fatty liver) and mortality. ? implies that its effect is not yet evaluated or probably lower.

## Data Availability

Not applicable.

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
