# Peer review of "The Roles of Carbohydrate Response Element Binding Protein in the Relationship between Carbohydrate Intake and Diseases"

_ijms, 2021, doi:10.3390/ijms222112058_

Round 1
Reviewer 1 Report
In this manuscript, author has provided the recent insights into role of carbohydrate response element binding protein (ChREBP) in glucose and lipid metabolism. The author suggested that amount, kind, and form of carbohydrate determine the glucose and lipid metabolism during disease progression and mortality. Although the theme of review is good, but the manuscript is written poorly. The organization of different sections and knowledge flow in the manuscript could have been better. There is no clear hypothesis and novelty. English language can be improved. There are many typos, grammatical errors, incorrect notations of gene names and numerous statements are not clear. If the paper is a review paper, then what kind of results has been discussed by the author. Author should follow IJMS guidelines for writing a review paper. Further comments are below:
- The current title is not good. It should be clear with precise scientific meaning. Therefore, I would suggest changing the title to attract more readers, and to maximize the accessibility of the review.
- Key words: Comma or semicolon?
- Introduction: This section is small and needs more effort from the author particularly last paragraph to develop more interest to the readers.
- Line 49-51: As there is only one author, ‘we’ can be replaced by ‘I’
- Line 57: ‘A recent study’ but author cited three references. Please change it to ‘studies’.
- Citations are not proper. e.g. Line 58, 65, 69 and many more, which year?
- There are many small paragraphs in the manuscript including 1.5 line paragraph (line 256). Few paragraphs can be merged.
- Figure legends are not enough informative. In fig 1, Y axis is missing.
Author Response
Response to reviewer 1’s comment
Q1: Although the theme of review is good, but the manuscript is written poorly. The organization of different sections and knowledge flow in the manuscript could have been better. There is no clear hypothesis and novelty. English language can be improved. There are many typos, grammatical errors, incorrect notations of gene names and numerous statements are not clear.
A1: Thank you very much for your suggestions. According to your suggestions, I thoroughly rewrote my review and changed several sections. That is why this response is not always point-by-point response to reviewer comments. English. Editing service provided by American Journal expert proofread my manuscript.
Q2: If the paper is a review paper, then what kind of results has been discussed by the author. Author should follow IJMS guidelines for writing a review paper.
A2: I checked author guidelines again.
Q3: The current title is not good. It should be clear with precise scientific meaning. Therefore, I would suggest changing the title to attract more readers, and to maximize the accessibility of the review.
A3: I corrected my title as follows: The Roles of Carbohydrate Response Element Binding Protein in the Relationship between Carbohydrate Intake and Diseases.
Q3: Keywords: Comma or semicolon?
A3: I wrote in the order of full spelling and its abbreviation.
Key words: Glucose; Carbohydrate response element binding protein, ChREBP; Fructose; Type 2 diabetes mellitus, T2DM; High-fructose corn syrup, HFCS; sugar-sweetened beverages, SSBs.
Q4: Introduction: This section is small and needs more effort from the author particularly last paragraph to develop more interest to the readers.
A4: I rewrote the introduction section.
Q5: Line 49-51: As there is only one author, ‘we’ can be replaced by ‘I’
A5: I corrected them.
Q6: Line 57: ‘A recent study’ but author cited three references. Please change it to ‘studies’.
Thank you very much for your comments. I corrected.
Q7: Citations are not proper. e.g. Line 58, 65, 69 and many more, which year?
A7: I checked them again and rewrote them.
Q8: There are many small paragraphs in the manuscript including 1.5 line paragraph (line 256). Few paragraphs can be merged.
A8: I corrected them
Q9: Figure legends are not enough informative. In fig 1, Y axis is missing.
A9: I corrected it.

Reviewer 2 Report
General opinion: This review article deals with an interesting and relevant aspect of dietary carbohydrate and energy utilization. However, it cannot be recommended for publication in its present form.
Although the main focus of the paper is the role of ChREBP, it devotes several pages to a shallow summary of the role of dietary carbohydrates in human health, a topic that has been profusely investigated and reviewed. In the opinion of this reviewer, these sections (1 to 2.2.3) add almost nothing to the basic knowledge in the field and should be condensed to provide only the background information needed to introduce the discussion on ChREBP and its physio/pathological role.
The author is encouraged to extend Sections 2.3 and the following, incorporating further details and discussion of the phenomena under consideration, As it stands now, it looks more like a collection of conclusions presented in the articles cited, with limited critical input from the review author.
Some specific comments:
Title: the term "amount" does not seem justified. The text contains almost nothing regarding quantitative intake data.
l. 33-34: sucrose and fructose cannot be "candidates for weight gain, obesity, and T2DM".
l. 36-37: where are those advanced glycation end products formed or accumulated?
l. 37-38: should glucose and fructose really be labeled as "toxic"?
l. 51-52: this suggests that original research data are being presented, which is not the case.
Heading 2.1: this is confusing since the section does not deal with carbohydrate intake in quantitative terms.
l. 65-67: the message here is not completely clear. Can it be reformulated?
l. 108: does the microbiota "digest" or ferment fructose?
l. 108-110: the message is not clear. Why is fructose consumption associated with irritable bowel syndrome?
l. 111-130: contain elements that seem somewhat contradictory. Please revise.
l. 177-179: these lines mention effects that do not seem directly derived from the previously (lines 168-176) mentioned facts.
Section 2.3: a more comprehensive molecular and physiological background on ChREBP is desirable.
l. 266-268: Does this mean that the inhibition of ChREBP modifies the gut microbiota mass? Please elaborate on this.
l. 271-273: the message is not clear. Please elaborate on this
l. 279-280: please elaborate on the relevance of this comment in relation to the role of ChREBP
l. 329-330: reference supporting this statement?
l.373-382: for being a review focusing on carbohydrates, the "future perspective" section ends with a strong emphasis on dietary protein metabolic effects. Please consider modulating these comments.
Figure 4: what is "sucrose insufficiency"?
The manuscript contains many abbreviations that may not be generally known. A list of abbreviations is thus desirable.
Please check the use of the abbreviation ChREPB. Sometimes it appears as Chrepb.
Author Response
Response to reviewer 2’s comment
Q1 This review article deals with an interesting and relevant aspect of dietary carbohydrate and energy utilization. However, it cannot be recommended for publication in its present form.
A1: Thank you for your suggestions. According to your suggestions, I thoroughly rewrote and reconstruct the sections. That is why the response to reviewer comments are not always point by point response.
Q2: Although the main focus of the paper is the role of ChREBP, it devotes several pages to a shallow summary of the role of dietary carbohydrates in human health, a topic that has been profusely investigated and reviewed. In the opinion of this reviewer, these sections (1 to 2.2.3) add almost nothing to the basic knowledge in the field and should be condensed to provide only the background information needed to introduce the discussion on ChREBP and its physio/pathological role.
A2: Thank you for your suggestions. According to your suggestions, I rewrote and reconstruct the sections.
Q3: The author is encouraged to extend Sections 2.3 and the following, incorporating further details and discussion of the phenomena under consideration, As it stands now, it looks more like a collection of conclusions presented in the articles cited, with limited critical input from the review author.
A3: According to your suggestions, I rewrote and reconstruct the sections. In the further perspection section, I wrote my opinions.
Some specific comments:
Q4:Title: the term "amount" does not seem justified. The text contains almost nothing regarding quantitative intake data.
A4 according to your suggestions, I corrected it.
Q5: l. 33-34: sucrose and fructose cannot be "candidates for weight gain, obesity, and T2DM".
A5: I rewrote it: fructose and sucrose induce fatty liver and body weight gain
Q6: l. 36-37: where are those advanced glycation end products formed or accumulated?
A6: AGE is included in foods and AGE is produced in the body, too. AGE is intracellularly produced and degraded. But, some proteins (extracellular matrix) with AGE are not digested and bound to Recptor for AGEs (RAGE), thereby inducing inflammation. This is not the main subject, so I'll omit it.
Q7: l. 37-38: should glucose and fructose really be labeled as "toxic"?
A7: I rewrote this.
Q8: l. 51-52: this suggests that original research data are being presented, which is not the case.
A8: I corrected it
Q9:Heading 2.1: this is confusing since the section does not deal with carbohydrate intake in quantitative terms.
A9: I corrected it.
Q10: l. 65-67: the message here is not completely clear. Can it be reformulated?
I corrected this.
Q11: l. 108: does the microbiota "digest" or ferment fructose?
A11: ferment is fine.
Q12:l. 108-110: the message is not clear. Why is fructose consumption associated with irritable bowel syndrome?
A12: Undigested fructose cause to ferment gut microbiota.
Q13: l. 111-130: contain elements that seem somewhat contradictory. Please revise.
A13: I rewrote it.
Q14: l. 177-179: these lines mention effects that do not seem directly derived from the previously (lines 168-176) mentioned facts.
Section 2.3: a more comprehensive molecular and physiological background on ChREBP is desirable.
A14: As the reviewer suggested, it is complicate to understand. I rewrote this section.
Q15: l. 266-268: Does this mean that the inhibition of ChREBP modifies the gut microbiota mass? Please elaborate on this.
A15: This implies that ChREBP inhibition cause to suppress fructose absorption and unabsorbed fructose then caused to ferement gut microbiota, thereby promoting irritable bowel syndrome.
Q16: l. 271-273: the message is not clear. Please elaborate on this
- 279-280: please elaborate on the relevance of this comment in relation to the role of ChREBP
- 329-330: reference supporting this statement?
A16: I rewrote it.
Q17: l.373-382: for being a review focusing on carbohydrates, the "future perspective" section ends with a strong emphasis on dietary protein metabolic effects. Please consider modulating these comments.
A17: I rewrote this section.
Q18:Figure 4: what is "sucrose insufficiency"?
A18: I changed sucrose insufficiency to sucrose intolerance.
Q19: The manuscript contains many abbreviations that may not be generally known. A list of abbreviations is thus desirable.
A19: Abbreviation section is not here.
Q20: Please check the use of the abbreviation ChREPB. Sometimes it appears as Chrepb.
A20: Chrebp and ChREBP are mouse and human gene symbol, respectively. ChREBP implies human and mouse protein symbol.
Round 2
Reviewer 1 Report
The manuscript has been improved significantly therefore can be considered for publication in its current form
Reviewer 2 Report
The manuscript has been substantially improved. It can be recommended for publication.